

# Modelling the interactive effects of viral presence and global warming on Baltic Sea ecosystem dynamics

Shubham Krishna⋆,[1], Victoria Peterson⋆,[1], Luisa Listmann[2], Jana Hinners[1]

[1]Helmholtz-Zentrum Hereon, Max-Planck-Straße 1, 21502 Geesthacht, Germany
[2]Institut of Marine Ecosystem and Fisheries Science, Universität Hamburg, Olbersweg 24, 22767 Hamburg
⋆ These authors contributed equally to this work

*Correspondence to*: Shubham Krishna (Shubham.krishna@hereon.de)

**Abstract**

Marine viruses have been identified as key players in biogeochemical cycles and in the termination of phytoplankton blooms;
however, most models of biogeochemical processes have yet to resolve viral dynamics. Here, we incorporate a viral component into a 1D ecosystem model for the Baltic Sea to explore the influence of viruses on ecosystem dynamics under current and future climatic conditions. Virus host interactions and zooplankton grazing were mechanistically described through size-based contact rates. The model demonstrated that the presence of viruses increased nutrient retention in the upper water column. This corresponded to a reduction in phytoplankton biomass, production of dead organic matter and transfer of biomass to higher
trophic levels. Viral presence played a key role in deeper water layers, near the thermocline. While warming alone reversed these trends, the combination of warming and viral presence enhanced the effect of viruses. Our results illustrate that marine ecosystem models need to incorporate viral dynamics to better predict system responses to climate change.

## 1 Introduction

In recent decades, viruses have been identified as major players in marine ecosystems, influencing nutrient availability, primary
production, and the transfer of biomass to higher trophic levels (Noble and Fuhrman, 1997; Suttle, 2007; Brussaard et al., 2008; Danovaro et al., 2011). Even though there is a growing body of literature that analyses phytoplankton-virus interactions using numerical models (Fuhrman, 1999; Weitz et al., 2015; Talmy et al., 2019; Flynn et al., 2021; Demory et al., 2021), the effect of viruses on biogeochemical dynamics still needs to be addressed in complex marine ecosystem models.

Viruses are estimated to be 10 times as abundant as prokaryotic cells and are the most abundant "life form" in the oceans
(Kirchman, 2008; Suttle, 2007). When viruses infect their phytoplankton hosts, organic matter recycling increases and matter transfer to higher trophic levels is reduced. This process is commonly referred to as the viral shunt, and can increase primary production by increasing nutrient availability in the upper water column (Fuhrman, 1999; Kirchman, 2008). Virus-mediated phytoplankton mortality was identified to be comparable to the importance of zooplankton grazing (Mojica et al., 2016). It is estimated that up to fifty percent of photosynthetically fixed carbon is released by viral infection (Fuhrman, 1999; Suttle, 2007;
Biggs et al., 2021). The role of viruses on the export of dead organic matter, the biological carbon pump, is less well understood.



The impact of viruses on their hosts is influenced by the relative viral abundance, host susceptibility to infection, as well as environmental factors (Baudoux and Brussaard, 2008; Mojica et al., 2016; Murray and Jackson, 1992; Suttle, 2007; Suttle and Chen, 1992). There has been evidence that the viral impact on phytoplankton communities is influenced by temperature and sunlight exposure. Previous studies on viral decay found that viral decay rates increased with sunlight exposure (Noble and

Fuhrman, 1997; Suttle and Chen, 1992; Lievens et al., 2022). There has also been evidence that the number of infected cells is higher in the absence of light (by night) than in the presence of light (Winter et al., 2004; Derelle et al., 2018), suggesting a connection between viral infection and irradiance. Additionally, temperature increases the rates at which viruses come in contact with their hosts (Murray and Jackson, 1992), enhancing viral lysis. Very high temperatures can instead lead to decreased viral lysis (Demory et al., 2021). Temperature as well as irradiance both vary over the span of the year and over the

depth of a water column. In addition, as the climate warms, changes in water temperature, nutrient availability, and light intensity may affect the interactions between viruses and phytoplankton (Bauer et al., 2013; Finke et al., 2017; Zhang et al., 2021). Thereby also biogeochemical dynamics such as the biomass transfer to higher trophic levels and the carbon export may be altered. To understand how these interactions play out in a given system, particularly with respect to climate change, it is important to resolve these interactions both across seasons and across the water column. A growing number of numerical

models address the interactions between phytoplankton and viruses. Weitz et al., (2015) evaluated the role of viruses in shaping community structure and ecosystem functioning using a multitrophic ecosystem model. Talmy et al., (2019) explored the impact of viruses and grazers on phytoplankton mortality in the California Current System. Demory et al., (2021) investigated the impact of temperature on viral infection of phytoplankton. While all these models expand our knowledge of the effects of phytoplankton-virus interactions, their formulation using zero-dimensional frameworks limits our understanding of how

phytoplankton-virus interactions are influenced across the water column. To our best knowledge, more complex one- to three-dimensional ecosystem models do not resolve phytoplankton-virus interactions yet. These interactions may be especially important in light of climate change, when temperature and light conditions change.

Here, we setup an idealized 1D ecosystem model that encapsulates the dynamics of viruses, their phytoplankton hosts, and zooplankton grazers during the Baltic Sea spring bloom. For this purpose, we developed an ecosystem model that considers

phytoplankton-virus and -zooplankton interactions using size-based contact rates. This design allows the model to be easily applied to different systems. We couple this model to the General Ocean Turbulence Model (GOTM) (Bolding, Karsten et al., 2021) utilizing the Framework for Aquatic Biogeochemical Models (FABM) (Bruggeman and Bolding, 2014) and parameterize the model with data on phytoplankton, viruses and zooplankton from the Kiel Bight (Listmann et al., unpublished). Using this model system, we explore the effect of viruses on Baltic Sea ecosystem dynamics under current and

future temperature conditions.



## 2 Material and Methods

### 2.1 Model description

We used the FABM interface (Bruggeman and Bolding, 2014) to couple GOTM with a biogeochemical-virus model (Bio-Vi). GOTM is a 1D physical model which calculates vertical turbulent fluxes of momentum, temperature, and salinity. Turbulence and tracer transport was described by Reynolds- averaged Navier–Stokes (RANS) equations in a rotating reference frame. A detailed description of GOTM is provided in Umlauf et al. (2005).

The biogeochemical model presented in this study has dedicated nutrient (DIN), phytoplankton, infected phytoplankton, virus,
zooplankton, and detritus compartments and resolves their respective dynamics and the exchange of fluxes between them (Fig. 1). The currency of the model is in terms of µmol N m$^{-3}$. We parameterized the model to represent the spring bloom phenology in the Kiel Bight. The coupled model was forced with the high-resolution environmental data obtained from the German Climate Computing Centre (https://www.wdc-climate.de/ui/).

### 2.1.1 Bio-Vi model

The processes for modelling matter transfer between the different compartments is lined out as a series of differential equations for nitrogen (Eq. 1), uninfected phytoplankton (Eq. 2), infected phytoplankton (Eq. 3), viruses (Eq. 4), zooplankton (Eq. 5), and detritus (Eq. 6). Description, values, and references for the parameters can be found in Table 1.

$$\frac{dN}{dt} = r * D + rzn * Z + rpn * P - P * \mu * i_{lim} * t_{lim} * n_{lim} \tag{1}$$

$$\frac{dP}{dt} = P * \mu * i_{lim} * t_{lim} * n_{lim} - c_z * P * \frac{Z}{n_Z} * p_c - c_V * P * \frac{V}{n_V} * p_a * p_i - m_P * P - rpn * P \tag{2}$$

$$\frac{dInf}{dt} = c_V * P * \frac{V}{n_V} * p_a * p_i - c_z * I * \frac{Z}{n_Z} * p_c - lys * I \tag{3}$$

$$\frac{dV}{dt} = lys * I * \frac{n_V}{n_P} * B_{size} - decay_{UV} * V - c_D * V * \frac{D}{n_D} * p_a \tag{4}$$

$$\frac{dZ}{dt} = e_z * c_z * p_c * \frac{Z}{n_Z} * (P + I) - m_Z * Z - rzn * Z \tag{5}$$

$$\frac{dD}{dt} = (1 - e_z) * c_z * p_c * \frac{Z}{n_Z} * (P + I) + \left(1 - \frac{n_V}{n_P} * B_{size}\right) * lys * I + m_P * P + m_Z * Z + decay_{UV} * V + c_D * V * \frac{D}{n_D} * p_a - r * D \tag{6}$$

### 2.1.2 Phytoplankton

The phytoplankton growth rate is assumed to be limited by nutrient availability (Eq. 7) (Monod, 1949), light (Eq. 8) (Webb et al., 1974) and temperature (Eq. 9) (Hinners et al., 2019). Thermal response parameters for Baltic Sea diatoms were mildly adjusted from (Warns, 2013) to match the spring bloom phenology of the Baltic Sea (Hjerne et al., 2019). The temperature





dependency of phytoplankton cell size (in terms of volume) in the model is adopted from Atkinson et al. (2003), see Eq. 10. Phytoplankton (diatom) volume ($vol_P$) was obtained from data from the Kiel bight (Fig. S6) and was converted to nitrogen

content using the volume-to-carbon ratio for diatoms, the C:N ratio, and particle radius (Menden-Deuer and Lessard, 2000), see Appendix Eq. S1.1.

$$n_{lim} = \frac{N}{kn + N} \tag{7}$$

$$i_{lim} = 1 - e^{\frac{-0.7 * par}{\mu}} \tag{8}$$

$$t_{lim} = e^{-\frac{(T - T_{opt})^2}{(T_1 - T_2 * sign(T - T_{opt}))^2}} \tag{9}$$

$$vol_P = vol_{15} + vol_{15} * 0.025 * (15 - T) \tag{10}$$

### 2.1.3 Encounter Rates for Grazing and Viral Lysis

One approach for modelling phytoplankton mortality due to zooplankton grazing or viral lysis is to use experimentally derived clearance rates. However, as system-specific literature on viral infections is still limited (Mojica et al., 2016), these interactions

were modelled using contact rates. The advantage of contact rates is that it allows for size to be the determining trait for interactions and helps in reducing the complexity of the model. Contact rates have been utilized by previous models to approximate the dynamics between viruses and their hosts, as well as their zooplankton grazers (Talmy et al., 2019; Flynn et al., 2021). Contact rates are a product of diffusion rates of the different components, their concentration, as well as additional traits such as swimming. Diffusion rates are calculated using the equation of Murray and Jackson (1992) and follows a standard

format for all particles (Eq. 11).

$$d_{particle} = \frac{kB * K}{6 * \pi * n * r_{particle}} \tag{11}$$

### 2.1.4 Viruses

Viral infection depends on contact (diffusion-based) between hosts and viral particles and their respective concentrations (Fig. 1). The contact rate $c_V$ is described in Eq. 12. Not all encounters between viruses and their hosts lead to the host becoming

infected (Eq. 13). For successful infection to occur viruses must first successfully adhere to their host cells ($p_a$) and then





successfully infect their host ($p_i$) (Flynn et al., 2021). Once infection occurs, the phytoplankton enters a separate pool of infected phytoplankton to account for a latent period between infection and lysis of the cell. The number of viral progenies released (burst size) is dependent on the growing conditions of the host. A host cell that is in poor growth conditions will produce fewer and more defective progeny than a host cell that is growing in optimum conditions (Mojica and Brussaard, 105   2014). This is described by the burst size limitation function ($B_{lim}$, see Eq. 14), which in combination with N content ratio of individual viral particles determines the quantity of nitrogen incorporated into viral progeny upon lysis, and the quantity released as cell fragments to the detritus (Eqs. 3 and 6).

$$c_V = 4 * \pi * (d_P + d_V) * (r_P + r_V) \tag{12}$$

$$Infection = c_V * P * \frac{V}{n_V} * p_a * p_i \tag{13}$$

$$B_{lim} = B_{min} + B_{max} * i_{lim} * t_{lim} * n_{lim} \tag{14}$$

Virus burst size parameters ($B_{min}$ and $B_{max}$) along with individual viral particle size are adopted from Flynn et al., (2021). Nitrogen content was estimated utilizing volume to C ratio for viral particles (Jover et al., 2014) and C:N ratio (see Eq. S1.2). 110   The adhesion of viral particles to non-host particles has been identified as a major cause of viral decay (Suttle, 2007). This is accounted for in our model through viral interactions with detritus particles and is parameterized similar to viral infection of phytoplankton (see Eq. 15). Ultra violet (UV) light enhanced decay of viruses is also considered in the model. Suttle and Chen (1992) found that wavelengths of light between 300-400nm have significant impact on viral decay rates. The approximated the decay in relation to water depth is described in Eq. 16, where k is the attenuation coefficient and z is the depth.


$$decay_{UV} = 1 - e^{\left(-0.1 * \frac{par}{decay_{surface}}\right)} \tag{15}$$

$$decay = decay_{surface} * e^{-kz} \tag{16}$$



### 2.1.5 Zooplankton

The grazing by zooplankton (Fig. 1) depends on diffusion-based contact between them and phytoplankton (Eq. 17), motility of grazers (Eq. 18) and a radius of detection (Talmy et al., 2019). It is assumed that only a fraction of phytoplankton will be

captured and grazed upon by zooplankton and the rest will remain free in the water column ($p_c$), see Eq. 19. Both uninfected and infected phytoplankton are grazed upon. Of the phytoplankton that are grazed upon a fraction is released to the detritus due to sloppy feeding or inefficient zooplankton grazing. Nitrogen content of zooplankton was calculated from size using equations from Broglio et al., (2003), which converts volume of zooplankton (mm3) to milligrams of N. This was then converted to units µmol N (Eq. S1.3).

$$c_Z = 4 * \pi * (d_P + d_Z) * (r_P + r_Z) + \pi * (r_P + r_Z * 3)^2 * s_Z \qquad (17)$$

$$s_Z = 0.01 * e^{(0.4 + 0.8 * \log(r_Z * 200))} \qquad (18)$$

$$Grazing = c_Z * P * \frac{Z}{n_Z} * p_c \qquad (19)$$

The zooplankton size parameters were estimated from sampling data from the Kiel Bight (Fig. S6). Nitrogen content ($n_z$) was calculated from size of zooplankton (Alcaraz et al., 2003), see Eq. S.3. The values for the standard zooplankton loss parameters, mortality and excretion rates, are the same as used in other studies for the same region (see Table 1).

### 2.2 Model runs

We analysed four scenarios of our model to investigate the role of viruses in the lower trophic levels of the Baltic Sea ecosystem

and how it would be impacted by climate warming. For the "Control" scenario, we forced the model with the present-day environmental conditions without considering viral dynamics. The "Control+Viruses" scenario accounted for virus-host interactions for the present period. The current trends in sea surface temperature increase in the Southern Baltic Sea and the North Sea are in the range of 0.25-0.35 °C per decade and an increase by 2°C is projected for the end of this century (Reusch et al., 2018). To study the effects of future climate warming on phytoplankton-virus dynamics, two additional model scenarios

were analyzed considering a 2°C increase in air temperature from the present, excluding ("Future") or including ("Future+Viruses") viral presence. For all four scenarios, the model was allowed to spin up for seven years and the results were then averaged over the following three years. In case of viral presence, the analysed phytoplankton biomass represents the sum of uninfected and infected phytoplankton.



## 3 Results

Here, we explored the role of marine viruses on biogeochemical dynamics in the Baltic Sea, coupling a biogeochemical-virus model (Bio-Vi) model to the one-dimensional water column model GOTM. With this model system, we explored the role of viruses under current and future temperature conditions.

### 3.1 Seasonal ecosystem dynamics

Our model is able to reproduce the observed seasonality in temperature and nutrient dynamics in the Baltic Sea (Lennartz et
al., 2014). From January to March, nutrients are uniformly distributed in the top 25 m due to convective mixing (Fig. 2a). Nutrients are completely depleted from the euphotic zone by the start of May and remain like this until autumn. A significant regeneration of DIN (concentrations going up to 2 mmol N m$^{-3}$) happens from July to October below 20 m depth. From November on the replenishment of DIN to the top layers starts to take place.

The spring bloom in our "Control" scenario starts in April (Fig. 2b) which is supported by observations from the Kiel Bight
(Hjerne et al., 2019). The overall phytoplankton biomass moreover matches observations regarding the order of magnitude (Hjerne et al., 2019). The "Control" scenario produces prolonged phytoplankton biomass at depth of 15m that persists until the end of September. The biomass of zooplankton follows the pattern of phytoplankton (Fig. 2c). Detritus accumulates along the bottom beginning in the spring and lasting until fall (Fig. 2d).

### 3.2 Impact of Viruses

In accordance with viral shunt dynamics, viral presence (Fig. 3) leads to an increase in nutrients during the summer months in the top 20 m and especially at depths between 15-20 m. There is a slight decrease in nutrients that coincides with the beginning of the bloom. Nutrient concentrations in the bottom 10 m are reduced during the late summer/early fall months. Phytoplankton biomass is slightly higher during the onset of the bloom but declined for the last half of the bloom with the greatest percent reduction in the sustained biomass at depth during the summer. The timing and the magnitude of the simulated phytoplankton
biomass of the "Control+Viruses" scenario qualitatively matches better with the observations than the scenario excluding viruses (Hjerne et al., 2019), suggesting that viruses might play an important role in phytoplankton bloom termination. Changes in zooplankton biomass resemble that of phytoplankton. The addition of viruses greatly reduces detritus accumulation at depth with minor increases in the upper water column.

Phytoplankton mortality due to grazing and viral infection are calculated as the percentage of phytoplankton biomass that is
lost due to zooplankton grazing or viral infection per day (Fig. 4). Grazing-related mortality is similar range in both control scenarios including or excluding viral presence (not shown). The maximum mortality by viral lysis is higher than that of grazing, but the time over which phytoplankton mortality by viruses can be detected is shorter than that caused by grazers. The simulated mortalities are in good agreement with collected data from the Kiel bight (Fig. S7), and similar in magnitude to rates present in the literature (Mojica et al., 2016).



**Interactive effect of Viruses and Warming**

To investigate the effect of projected warming on ecosystem dynamics, we forced our model with an air temperature increased by 2°C (Reusch et al., 2018). On average, our model predicts an increase by 1.5°C in winter water temperature (Fig. S2), the stratification starts one month earlier and lasts one month longer compared to the "Control" scenario. The maximum increase of 2°C is reached in the upper productive layer during summer.

To evaluate seasonal changes between all four scenarios, the biomass of each compartment was integrated over depth for the span of year (Fig. 5). Depth resolved illustrations of all scenarios are visualized in Fig. S3-S5. Warming alone ("Future" scenario) leads to an earlier onset of the spring bloom and the highest phytoplankton biomass of all four scenarios (Fig. 5b). Thus, enhanced growth stimulated by higher temperatures seems to play a bigger role than stronger stratification causing a lack of nutrient supply from deeper water layers. In consequence, higher primary production causes lower summer nutrient conditions, an enhanced zooplankton biomass and enhanced detritus production.

As expected, the presence of viruses leads to increased nutrient concentrations during spring and summer (Fig. 5a). Viral presence moreover reduces the bloom duration and biomass levels observed over the summer months for both temperature scenarios (Fig. 5b). These effects are strongest though in the "Future+Viruses" scenario, illustrated also by the higher viral biomass in the "Future+Viruses" scenario. The earlier onset of the phytoplankton bloom under warm conditions moreover causes an earlier increase of virus biomass. Thus, while warming alone causes increased primary production, the combined effect of warming and viral presence enhances the negative effect of viral presence on primary production. These trends are also mimicked in the biomass and seasonal progression of zooplankton (Fig. 5c) and detritus (Fig. 5e). Viral presence and particularly viral presence under warm conditions causes a lower zooplankton biomass and lower production of detritus.

**4 Discussion**

In this study we explored the role of viruses on Baltic Sea spring bloom dynamics under current and future climatic conditions. Our simulations show that viruses lead to a shorter, weaker phytoplankton bloom, as well as decreased zooplankton and detritus biomass, whereas nutrient recycling is increased. Climate warming enhances these effects, nearly doubling virus biomass (Fig. 6).

While recent modelling efforts focused on describing host virus interactions in zero-dimensional frameworks (Talmy et al., 2019; Weitz et al., 2015; Demory et al., 2021; Flynn et al., 2021), there is a lack of research exploring virus-host dynamics in more complex ecosystem models. To the best of our knowledge, this is the first one-dimensional ecosystem model explicitly simulating virus-host dynamics. Since this model is constructed using a contact-based, size-dependent approach, it can be easily applied to different virus-phytoplankton systems other than the Baltic Sea example explored here. Our results support those of empirical research suggesting that viruses play a key role in ecosystem dynamics, particularly in the light of climate change.



## 4.1 Role of viruses in the Baltic Sea food web

The results presented in this study show how viruses can affect primary production, higher trophic levels, nutrient cycling and the export of particulate matter in the Baltic Sea across the water column. It is well established, that virus lysate is composed of labile substances (rich in free and combined amino acids) that are used by bacteria to mediate the transfer of nutrients from particulate to dissolved form. This process is called the "viral shunt" (Wilhelm and Suttle, 1999; Middelboe et al., 2003; Middelboe and Jørgensen, 2006). Our model simulations show this viral shunt. The addition of viruses leads to an increase in DIN concentrations and a decrease in phytoplankton and detritus biomass (Fig 6). The nutrient regeneration is particularly strong in deeper layers (15 to 20 m) during stratified conditions. This increase in nutrients can be explained by the enhanced viral lysis around the thermocline. Water temperatures and light conditions around the thermocline are high enough for phytoplankton growth, whereas the lower light intensity in comparison to surface waters is beneficial for the light-sensitive viruses (Noble and Fuhrman, 1997; Suttle and Chen, 1992). As a result, the viral shunt is enhanced, leading to the production of inorganic nutrients. Our results of a higher viral activity close to the thermocline are in accordance with previous empirical findings (Cochlan et al., 1993; Weinbauer et al., 1995). However, while viral presence causes a regeneration of nutrients, it does not lead to enhanced primary production in our control simulations (Fig 6). This is explained by the summer stratification, which limits the supply of nutrients in the top sunlit layers.

The termination of the Baltic Sea spring bloom has been attributed to different factors, such as the formation of resting stages (Hinners et al., 2019), grazing pressure (Wasmund et al., 2019), or simply the exhaustion of nutrients (Tamminen and Andersen, 2007). Our simulations suggest that the presence of viruses leads to a significant decline in phytoplankton biomass (reduction by ~27%, Fig. 6), particularly during the period from spring to early summer. We therefore conclude that viruses may play a key role in the termination of the Baltic Sea spring bloom. This is in agreement with previous research that suggests viruses are a significant source of phytoplankton mortality in the oceans (Suttle, 2000; Danovaro et al., 2011; Wirtz, 2019).

By increasing phytoplankton mortality, viruses can dampen the flow of matter and energy fluxes to higher trophic levels. In our simulations, this is reflected in the decline in zooplankton concentrations when viruses are present, which agrees with previous modelling efforts (Weitz et al., 2015).

## 4.2 Interactive effects of viruses and future warming on phytoplankton and export dynamics

Marine ecosystems are impacted by a variety of stressors acting in conjunction. Lately, the importance of studying and understanding the interactive effects of multiple stressors on physio-ecological traits in marine ecosystems has received more attention (Adams, 2005; Crain et al., 2008; Andersen et al., 2020; Gissi et al., 2021). A combined response of two stressors is defined as synergistic when it is higher in magnitude than the sum of individual responses to each of the two stressors. It is antagonistic when the combined response is smaller than the addition of individual responses (Crain et al., 2008). Marine viruses are known to interact actively with climate change and are able to influence oceans' feedback to the atmosphere (Danovaro et al., 2011). In this study, we consider viruses and warming both as potential stressors on phytoplankton growth.





The comparison of the results of our four model experiments ("Control", "Control + Viruses", "Future", "Future + Viruses") allows us to interpret how viruses and climate warming could interactively affect phytoplankton and export dynamics in the Baltic Sea. We focus on these two aspects due to their importance for carbon cycling. The viral-driven mortality of
phytoplankton under control conditions accounts for a loss of total biomass of primary producers by ~27% compared to simulation excluding viral presence (Fig. 6). Conversely, future climate warming without considering viral presence leads to an increase in phytoplankton biomass by ~14%. This positive influence of warming on primary production is mainly due to reduced winter mixing increasing water transparency. Higher water transparency improves light conditions in the productive
layer allowing for stronger primary production. In contrast, our simulations including warming and viral presence predict a decrease by ~33% in total phytoplankton. Thus, although future warming alone has a positive impact on phytoplankton, viruses and warming combined have a synergistic negative effect on the biomass of autotrophs. This synergistic negative effect is caused by the increased phytoplankton mortality due to warming-stimulated viral activity. This finding supports previous empirical and modelling efforts on the effects of viral presence under warmed conditions (Murray and Jackson, 1992; Mojica
et al., 2016; Biggs et al., 2021; Piedade et al., 2018). Demory et al. (2021) report though that very high temperature conditions corresponding to tropical regions can also reduce viral lysis.

Our results show that climate warming and viral presence have contrasting effects on detritus formation when they act alone. Since our model is tracking the pathway of nitrogen, we can make presumptions for the export of carbon assuming the Redfield ratio. While warming alone increases carbon export by ~9%, viral presence reduces carbon export by ~17% under current
environmental conditions. This effect is strengthened for viral presence under warm conditions (reduction by ~30%). These statements need to be taken with caution though since the stoichiometry of detritus is highly variable and can differ considerably from the phytoplankton's cellular composition (Frangoulis et al., 2004). Nevertheless, a virus-mediated decrease in organic matter export under warm or stratified conditions, respectively, is consistent with previous experimental and conceptual work (Lawrence and Suttle, 2004; Piedade et al., 2018). Moreover, viral lysis can alter the composition of marine
snow aggregates and influence carbon pump efficiency. The outcome of these virus-mediated activities remain uncertain though (Weinbauer et al., 2011).

### 4.3 Model biases and outlook

The focus of our study lies on the spring bloom dynamics in the Baltic Sea. As a shortcoming of this focus, our model shows unrealistically high summer nutrient concentration close to the sea surface in the "Future + Viruses" scenario. For this reason,
we repeated all simulations using a two-group model, including a second phytoplankton compartment, representing picophytoplankton from the Baltic Sea with a higher temperature range for growth (Santelia et al., unpublished). While surface water nutrients become exhausted during summer in this two-group model, the overall distribution of biomass across the different compartments shows the same pattern for the four scenarios (Fig. S8, S9).

Competition between species is very influential on population dynamics, niche formation, and bloom timing, with most marine
systems typically demonstrating a seasonal progression in species composition (Sommer and Lengfellner, 2008; Hjerne et al.,



2019). The response to temperature increase has also been shown to be influenced by the composition of the community (Striebel et al., 2016). Following the "Kill-the-winner hypothesis", viruses rely on high host density to propagate and as such are more likely to affect fast growing or most abundant species. This tendency influences biodiversity by allowing other slower growing, less successful species to persist (Suttle, 2007). For this reason, the impact of viruses on phytoplankton growth

dynamics should be further explored within a more complex framework that includes competition dynamics of multiple species.

Moreover, our model system can be extended by incorporating adaptive responses of phytoplankton and their viruses. Both viruses and phytoplankton have short generation times, which give them the ability to adapt to changing environmental conditions. Phytoplankton have been shown to be able to adapt quickly to changes in temperature (Schaum et al., 2017;

Listmann et al., 2016).   Furthermore, phytoplankton can express genetic variability in the susceptibility to infection (Suttle, 2007). While the model does consider a low probability of infection following successful contact, high infection rates present a situation in which infection-resistant lines of phytoplankton are highly selected for and can increase in abundance within the population. As such, it is likely that not all phytoplankton are susceptible to viral infection during a bloom event and that the proportion of phytoplankton that are susceptible decreases throughout the bloom, when resistant phytoplankton become

dominant. Additionally, viruses themselves have been shown to manipulate the genome of their hosts and provide new genetic material (Rohwer and Thurber, 2009). In this way they can alter their host's thermal performance curve (Padfield et al., 2020). Viruses can thus represent an important evolutionary driver in phytoplankton communities.

### Conclusions

In this study, we propose an idealized model setup to study contact-based, size-dependent virus-phytoplankton interactions in

the Baltic Sea. To our best knowledge, this is the first one-dimensional ecosystem model resolving the effects of viral presence for the region of Baltic Sea. Our simulations demonstrate that viruses represent a key driver of ecosystem dynamics, which will play an even more important role as climate warming continues, with cascading effects on phytoplankton bloom duration, biomass transfer to higher trophic levels, and carbon export. Since our model is designed using size-based, contact interactions, it can easily be applied to other phytoplankton-virus systems in future research efforts.

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



## Tables and Figures

**Table 1. Description of model parameters with their units.**

| Parameter | Description | Unit | Value | Reference |
|---|---|---|---|---|
| $N$ | concentration Nutrients | µmol N m$^{-3}$ | - | |
| $P$ | concentration Phytoplankton | µmol N m$^{-3}$ | - | |
| $Z$ | concentration Zooplankton | µmol N m$^{-3}$ | - | |
| $I$ | concentration Infected | µmol N m$^{-3}$ | - | |
| $V$ | concentration Viruses | µmol N m$^{-3}$ | - | |
| $D$ | concentration Detritus | µmol N m$^{-3}$ | - | |
| $p0$ | background P concentration | µmol N m$^{-3}$ | 0.00001 | |
| $z0$ | background Z concentration | µmol N m$^{-3}$ | 0.00001 | |
| $kc$ | specific light extinction of P and D | m$^2$ µmol$^{-1}$ | 3.0e-5 | 1 |
| $i_{min}$ | min light intensity in euphotic zone | W m$^{-2}$ | 25.0 | 1 |
| $r$ | remineralization rate | d$^{-1}$ | 0.005 | 1* |
| $kn$ | half saturation constant | | 300.0 | 1 |
| $rpn$ | P excretion | d$^{-1}$ | 0.01 | 1 |
| $\mu$ | maximum growth rate P | d$^{-1}$ | 1.0 | 1 |
| $m_P$ | P mortality euphotic – below euphotic | d$^{-1}$ | 0.01-0.05 | 1 |
| $lys$ | lysis rate | d$^{-1}$ | 1.0 | ** |
| $e_Z$ | efficiency rate Z | - | 0.5 | 8* |
| $m_Z$ | mortality rate Z | d$^{-1}$ | 0.2 | 9 |
| $rzn$ | Z excretion | d$^{-1}$ | 0.01 | 1 |
| $r_Z$ | radius Z | m | 100e-6 | 6 |
| $decay_{surface}$ | decay rate of viruses at surface | d$^{-1}$ | 3.0 | 7* |
| $r_V$ | radius V | m | 4.25e-8 | 6 |
| $vol_P$ | volume P at 15°C | µm$^3$ | 11.0 | 6 |
| $vol_Z$ | volume Z at 15°C | µm$^3$ | 4.19e6 | 6 |
| $r_D$ | radius D | m | 4.0e-6 | ** |
| $n_V$ | nitrogen content V particle | µmol N m$^{-3}$ | 2.5e-12 | 2,5 |
| $n_D$ | nitrogen content D particle | µmol N m$^{-3}$ | 2.87e-7 | ** |
| $kB$ | Boltzman constant | g m$^2$ s$^{-2}$ K$^{-1}$ | 1.23e-23 | 2,3,4 |
| $n$ | dynamic viscosity of medium | g m$^2$ s$^{-1}$ | 9.9e-4 | 2,3,4 |
| $p_a$ | probability of adhesion | - | 0.01 | 2* |
| $p_i$ | probability of infection | - | 0.003 | 2* |
| $p_c$ | probability of capture | - | 0.07 | 2* |
| $B_{min}$ | minimum burst size | particles | 50 | 2 |
| $B_{max}$ | maximum burst size | particles | 450 | 2 |
| $T_{opt}$ | optimum temperature P | °C | 10 | ** |
| $T_1$ | width parameter of reaction norm | °C | 1.0 | ** |
| $T_2$ | width parameter of reaction norm | °C | 8.0 | ** |
| $w_p$ | vertical velocity of P | m d$^{-1}$ | -0.1 | 1* |
| $w_d$ | vertical velocity of D | m d$^{-1}$ | -5.0 | 1 |

Burchard et al., 2005[1]; Flynn et al., 2021[2]; Talmy et al., 2019[3]; Murray & Jackson 1992[4]; Jover et al., 2014[5]; Listmann et al., unpublished[6]; Suttle & Chen 1992[7]; Weitz et al., 2015[8]; Maar et al., 2014[9]
*values visually adjusted; **values approximated




**Figure 1: Schematic of the biogeochemical model. Top: Depicted pathways (arrows and purple labels) show fluxes of nitrogen between compartments (DIN, phytoplankton, infected phytoplankton, zooplankton, viruses, and detritus); colored spheres indicate**





**factors that limit or influence the rate at which these interactions occur. Middle and bottom: Schematic of contact-based viral infection and zooplankton grazing.**

**a) Nutrients**

**b) Phytoplankton**

**c) Zooplankton**

**d) Detritus**

**Figure 2: Concentrations of a) nutrients, b) phytoplankton, c) zooplankton, and d) detritus over seasons and 30 m depth for the "Control" scenario.**






**Figure 3: Change in concentrations for a) nutrients, b) phytoplankton, c) zooplankton, and d) detritus between the "Control and "Control+Viruses" scenario. Blue tones indicate an increase in concentration when viruses are present. Pink tones indicate a decrease in concentration when viruses are present.**





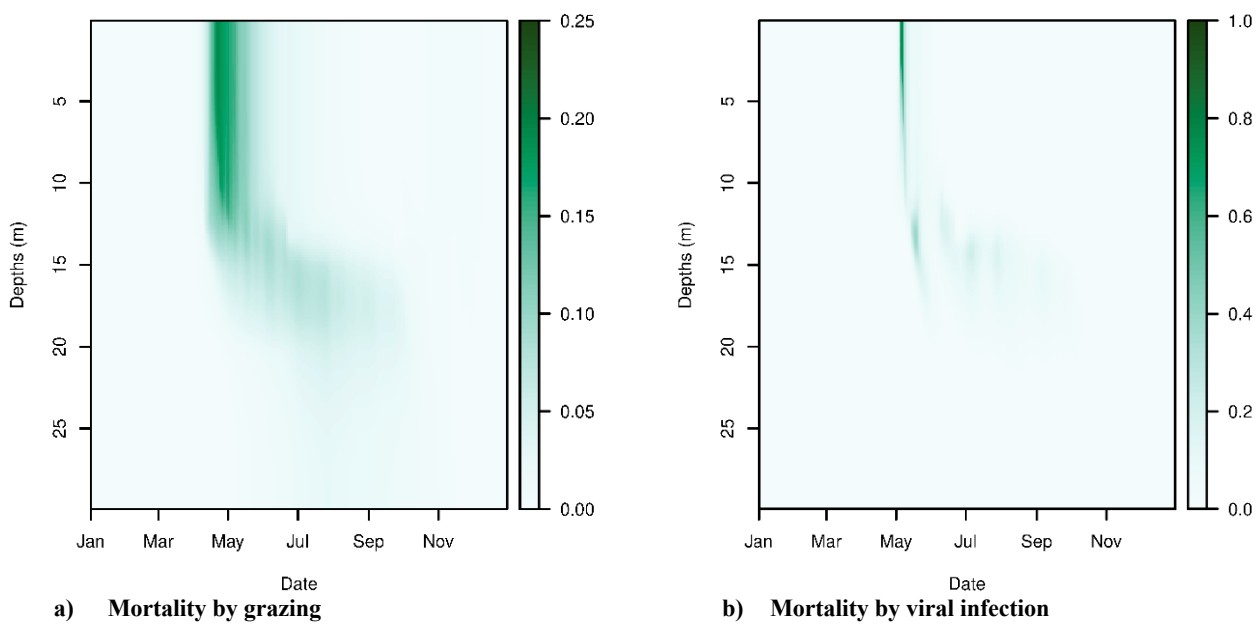

a)    **Mortality by grazing**                          b)    **Mortality by viral infection**

**Figure 4: Phytoplankton mortality for the "Control+Viruses" scenario caused by zooplankton grazing (a) and viral infection (b), resolved over seasons and depth.**






**Figure 5: Seasonal dynamics of biomass of all compartments accumulated over depth. Scenarios are illustrated using different colours.**






Figure 6: Annual biomass of all compartments integrated over depth and seasons. Scenarios are illustrated using different colours.






**Author contribution**

**Shubham Krishna:** Conceptualization, Methodology, Software, Writing - original draft, Review & editing

**Victoria Peterson:** Methodology, Software, Data curation, Analyses, Review & editing

**Luisa Listmann:** Data curation, Review & editing.

**Janna Hinners:** Conceptualization, Methodology, Review & editing.

**Competing interests**

None.

**Code/Data availability**

Will be made available upon request. Please write to shubham.krishna@hereon.de.
