# Peer review of "Modelling the interactive effects of viral presence and global warming on Baltic Sea ecosystem dynamics"

_Biogeosciences, 2022_

## Author Comment (AC1)

Summary

A 1D ecosystem model parameterized for the Baltic Sea is used to assess impacts of viral infection on biogeochemical processes (primary productivity, carbon export). A set of experiments are performed, with and without a combination of viruses and warming conditions. The study finds an interactive effect of warming and temperature on primary productivity, and carbon export.

Main comments

The study is generally clearly written, and the results are novel and interesting. The findings have clear implications for our understanding of viral impacts on ocean biogeochemistry in a future ocean. I did find a few areas where results were presented confusingly. There are also some formatting issues that may need to be addressed, and one minor query about model structure. These are all outlined below.

*- We thank the reviewer for the positive comments on our study and suggestions to improve the manuscript. Below, we address all the specific comments.*

Specific comments

General comment: the manuscript says 'code available on request'. It would be nice to make this available in such a way that others might be able to reproduce the findings.

*- We fully agree and we will provide the link to the data repository in the revised version of the manuscript.*

Page 2 line 50: "to our best knowledge…" a recent study assessed viral impacts on ocean biogeochemistry in a 1D setting at two ocean sites (Xie and Zhang 2022). I recommend *including this citation and outlining how the present study differs.*

*- Thank you for pointing us at this study, which we have missed. We will include this study in our citations and set it in context with ours.*

Page 6, equation 19: it looks like this is linear w.r.t. P? So, equivalent to Holling I, aka mass action? This implies the rate of grazing is unbounded, such that at very high P concentration, it can become quite large. It's a little unusual not to bound the rate of grazing with Holling II, Michaelis-Menten, or similar (e.g., Gentleman et al. 2003). Would be nice to see if including this makes a big difference to the results.

*- We decided to model both the viral lysis and grazing as contact-based rates. We chose this design following the study by Talmy et al., 2019. We chose this description to model both viral lysis and grazing dynamics in the same manner. While we agree that the unbounded grazing could potentially overestimate grazing under high phytoplankton and zooplankton concentrations, we do not see any sign for this in our results. Even if we exclude viral presence, zooplankton grazing does not lead to an inhibition or a rapid termination of the phytoplankton bloom. In fact, the phytoplankton mortality caused by zooplankton grazing in our simulations is in the same order of magnitude as laboratory experiments, as described in L168. However, we are happy to discuss the potential*

*overestimation of zooplankton grazing if the model is applied to other systems in the revised version of the manuscript.*

Page 7 line 159-161: "…qualitatively matches…" I'm fine with this sort of qualitative comparison, but am I right in saying that for me to evaluate the consistency, I need to access Hjerne et al. 2019, and determine which of their data is being referred to? This seems like a heavy lift, and I suspect most readers will not make the effort. Can these data be recreated here, as you have with the Mojica 2016 data (Figure S7).

*- We do not have the raw data from Hjerne et al. 2019, so we are not able to recreate their results. We compared our simulations to the visualization in their study (Fig. 2). However, we can plot our supplementary data (S6) in a similar way to the data by Hjerne et al to allow for a better comparison with our model simulations. For clarification, the data in Fig S7 is our own collected data on phytoplankton mortalities caused by viral lysis and grazing, not data from Mojica et al. 2016.*

Page 7 line 166: "The maximum mortality…shorter than that caused by grazers". I was curious as to why this is. I couldn't find it explained in the discussion. My apologies if I missed it. If an explanation hasn't been provided, please consider including one.

*- Thank you for pointing us at this, we are happy to discuss this in more detail in the revised version.*

*- These dynamics are caused by the contact-based description of our viral lysis and grazing. While viruses in our simulations with a very small diameter of 43 nm have a higher maximal clearance rate, zooplankton with their large size are more competitive at low zooplankton and virus abundances.*

*- In early spring, as long as zooplankton and viruses are rare, zooplankton graze more effectively due to their larger size resulting in a higher contact-based grazing rate. For this reason, zooplankton abundances (and their grazing mortality) increase earlier in the year. Once viruses become more abundant though, their rapid increase in numbers outcompetes zooplankton, leading to a drastic increase in virus-mediated mortality and a rapid decline of the phytoplankton spring bloom.*

Page 7 line 168: "(Fig S7)". Slightly pedantic on my part, but I expect the figure numbers to appear sequentially in the manuscript. This is the first supplemental figure and it goes straight to figure S7. Where are figures S1-S6 discussed? Are these discussed in the main text? Please make sure that all supplemental figures are discussed in sequence in the main text.

*- We will make sure figures S1-S6 are discussed in the revised version of the manuscript and that all figures are mentioned in chronological order.*

Page 8 line 176-180: "depth resolved … detritus production" I don't understand the reasoning here. What does it mean to say that "higher temperatures seem to play a bigger role than stratification"? Surely temperature is mechanistically linked with stratification? Do you mean to say that, the effect of temperature on biological rates has a stronger impact than the effect of temperature on stratification? If so, how can you conclude this? It's not at all clear to me what is being said here.

*- Thanks for raising this question, we will clarify this in the revised version. Warming can have two contradicting effects on phytoplankton. On the one hand, it can increase primary production due to the warming-driven stimulation of growth. One the other hand, warming can cause a stronger stratification of the water column, prohibiting the supply of nutrients from deeper water layers, thereby limiting primary production. Because the net effect of warming on phytoplankton biomass is positive in our simulations, we conclude that the stimulation of growth by temperature increase plays a larger role than the nutrient limitation resulting from increased stratification.*

Page 8 line 184-185: "The earlier onset …. Causes an earlier increase of virus biomass". I can't seem to find an explanation here or in the discussion for why this is the case. Apologies if I missed it. If an explanation isn't included, please provide one.

*- The earlier onset of the spring bloom is caused by the earlier increase in water temperature in the "Future" and "Future+Viruses" scenarios. We will clarify this in the revised version.*

Page 8 line 196: "to our best knowledge" as above, there is one study with viruses in 1D (Xie and Zhang 2022). Please cite and explain how the present study differs.

*- As mentioned above, we will set our study in context to this study in the revised version.*

References

Gentleman, W., A. Leising, B. Frost, S. Strom, and J. Murray. 2003. Functional responses for zooplankton feeding on multiple resources: A review of assumptions and biological dynamics. Deep. Res. Part II Top. Stud. Oceanogr. **50**: 2847–2875.

Xie, L., and R. Zhang. 2022. Assessment of Explicit Representation of Dynamic Viral. Viruses **14**: 1–21.

*- Hjerne, O., Hajdu, S., Larsson, U., Downing, A. S., & Winder, M. (2019). Climate driven changes in timing, composition and magnitude of the Baltic Sea phytoplankton spring bloom. Frontiers in Marine Science, 6, 482.*

*- Talmy, D., Beckett, S. J., Taniguchi, D. A., Brussaard, C. P., Weitz, J. S., & Follows, M. J. (2019). An empirical model of carbon flow through marine viruses and microzooplankton grazers. Environmental Microbiology, 21(6), 2171-2181.*

---

## Author Comment (AC2)

This work tackles parts of an important subject of relevance to the journal. However, it does so in such a simplistic way that the conclusions are either obvious (viruses impact plankton production) or unbelievable (results are based on a model describing only one phytoplankton type, so there is no competition between phytoplankton that are more or less impacted by their own species-specific virus and/or by the likely selection zooplankton grazing). For some reason the authors do not seem to be aware of Flynn et al. 2022, which goes into various of the matters considered here and shows the critical importance of using a multi-species model. Here, the authors have actually used a 2-phytoplankton variant of their approach, but this is mentioned rather in passing in Discussion. If the whole work had been conducted using that more complex model then the work would have been on a much firmer grounding.

We thank the reviewer for the helpful and detailed comments on our manuscript.

Regarding the design and chosen complexity of our model system:

We designed our model based on state-of-the-art descriptions of phytoplankton-virus dynamics (Talmy et al. 2019, Flynn et al., 2021). Similar to those models, we described the spring bloom phytoplankton as a single compartment. The novelty of our model lies in resolving phytoplankton-virus interaction dynamics in a 1-dimensional water column & ecosystem framework. We thank the reviewer for pointing us at the study by Flynn et al, 2022, which we clearly missed to cite in our manuscript. We see a lot of potential for future research using multi-species approaches as described in Flynn et al. 2022. However, we chose the description of phytoplankton using one "bulk" compartment to be able to compare our model with available data on the effects of viral lysis and grazing on the bulk of Baltic Sea spring bloom phytoplankton.

Since we are aware that our model description cannot fully represent the complexity of the Baltic Sea ecosystem, we tested a more complex version of the model with two phytoplankton (size) groups, their two respective viruses and two zooplankton (size) groups. We will describe the design of the complex model in more detail in the revised version.

However, by increasing the model complexity, the uncertainty in model estimates increases as well (Fulton et al., 2003). Since we lack data on species-resolved viral lysis and grazing, we cannot accurately constrain all physiological parameters for the more complex model setup. To strike a balance between complexity and a reasonable description of food web dynamics, we decided to focus our manuscript on the initial NPZD-V model design. This being said, the results of our complex model with two phytoplankton groups agree with the results of our initial model. For this reason, we think that our results are not necessarily dependent on an increase in model complexity. We will make sure to discuss potential shortcomings of our chosen model design in more detail in the revised version.

DETAILED COMMENTS

L12 Virus-host dynamics are highly specific; the specificity of this interaction here needs to be made very clear in the abstract.

L12 We will make sure to address the specificity of virus-host dynamics in the revised version.

L16 How did this warming interaction come about?

L16 The interaction of warming and viruses is caused by the dependence of the viral burst size on the growth conditions of the host. Under nutrient-replete conditions, a higher temperature will lead to a higher growth rate (unless the optimum temperature is exceeded), causing an increase in viral progeny. This is described in L 102ff.

L22 Is there a specific reason for not referencing Flynn et al. 2022 - it seems to have rather a lot in common with this submission.

L22 We thank the reviewer for pointing us to the recent study by Flynn et al., 2022, which we missed to refer to. We will relate it to our study in the revised version.

L26 Such an increase in primary production is not assured, and depends on the timing of events; these are matters for which models can help.

L26 We will describe the viral shunt dynamics with more caution in the revised version.

L28 It is very important to indicate early on that virus induced mortality is very different to that induced by zooplankton.

L28 We will point at the difference between virus induced mortality and zooplankton grazing in the revised version.

L69 It is very important to make it clear how many phytoplankton-virus couples are considered here - from what I can see there is just the one, implying that the Baltic has only one phytoplankton species with its virus and zooplankton. That is surely too much of a simplification. When a virus attacks its host, we must expect other phytoplankton to come to dominance. Whether they are suitable prey for the zooplankton is another important matter.

L69 We would like to refer you to our explanation regarding the chosen complexity of our model above. We will make sure to state the complexity of our model more accurately here.

L83 Cell size is affected by factors other than temperature, and certainly the species composition (and thence the specificity of any virus attacks) will be affected during successions.

L83 While there are other (more-complex) factors influencing cell size, e.g., grazer presence (Flynn et al., 2022), we modelled changes in cell size based on temperature alone, since this is a well-established correlation (Atkinson et al. 2003) and we investigate the effects of temperature on the dynamics of the ecosystem.

L134 I really do not see how such runs can possibly be related to reality. What happens depends as much on how uninfected species behave as it depends on that of virus-affected species.

L134 Please see our explanation regarding the chosen complexity of our model above.

L153 Most of what is released when phytoplankton burst would contribute to the DOM pool (as per L203), not to detritus. This does not appear to have been modelled, and neither is the activity of bacteria (and their grazers) that would be stimulated by such an event.

L153 The currency of our model is nitrogen. We fully agree with the reviewer that the exudation from phytoplankton contributes to the dissolved organic matter (DOM) pool. However, studies have shown differential remineralization of dissolved inorganic carbon, nitrogen and phosphorus (DOC, DON and DOP) to their inorganic counterparts and have reported that DON is least preferentially remineralized, and thereby not contributing significantly to the pool of dissolved inorganic nitrogen (DIN) in high latitude oceans (Sigman and Casciotti 2001, Wetz et al., 2007, Letscher et al., 2015). For this reason, we do not explicitly model the release of DON by phytoplankton. We do account for microbial remediated regeneration of nutrients through remineralization of detritus. Our detritus compartment comprises both dissolved and particulate organic matter (nitrogen). We will clarify this in the revised version of the manuscript.

L213 Virus presence alone cannot lead to a regeneration of nutrients (by which I assume you mean inorganic nutrient). I do not see how, at least in the system modelled, virus attack could ever promote primary production. Can it?

L213 According to the "viral shunt" hypothesis (Poulton et al., 2021), viral lysis leads to the production of dissolved organic matter (simulated as part of the detritus in our model), which in turn becomes remineralised leading to the increase in inorganic nutrient levels. Nutrient levels can then potentially favor increased primary production. We will describe this in more detail in the revised version of this manuscript.

L220 This model really cannot support such a claim; to do so it needs to describe the biodiversity of the plankton, and the allied specificity of viruses on components of the community.

L220 We would like to refer to our statement above on the chosen complexity of our model. Based on our model results, we still believe that our statement that viruses can play an important role in the termination of the spring bloom in the Baltic Sea is valid. However, we will discuss the possible biases of our chosen model design in the revised version.

L231 What does this 'interact actively' term mean? Viruses cannot do anything alone; they reply on the success of their host, and thence on many factors. This statement seems rather exaggerated.

L231 Danovaro et al., 2011 describe that "marine viruses interact actively with the present climate change". We describe our definition of the interaction of two stressors in the previous lines in the manuscript. We regard viruses as a stressor. For this reason, we investigated the interactive effects of two stressors (viruses and climate change). We will clarify this in the revised version.

L246 While this paragraph is interesting, and begs additional questions, I fear that the model is far too simple to make generalised claims like this.

L246 Please see our explanation regarding the chosen complexity of our model above. We discuss the biases of our model in the discussion section L258ff.

L261 How was the zooplankton configured to handle this additional prey item?

L261 In the more complex model version, we included two phytoplankton groups (diatoms and picophytoplankton), their respective viruses and zooplankton grazers. We will describe this model version in more detail in the revised version.

L268 This is incorrect. They only impact their specific host, and the ramifications from the different host-virus interactions with competition appears (from Flynn et al. 2022) to be complex and profound.

L268 We find the model simulations by Flynn et al. 2022 on the role of viruses of species dominance very compelling and are happy to discuss the ramifications for the Kill-the-winner hypothesis in the revised version.

Fig.1 That this is operated within a 1D scenario based on a real hydrodynamic scenario makes it no more representative than models operated in theoretical scenarios. The problem here is that the trophic setup is far, far, too simplistic. Viruses would only impact their own host; the idea that all phytoplankton would be impacted simultaneously in nature is not plausible. There is no bacterial activity simulated here (with or without their own viruses). Excretion of DIN by phytoplankton? What types of zooplankton are these (I assume from the 'sloppy feeding' term they are metazoan?).

Fig. 1. The aim of our study was to study ecosystem dynamics caused by viral lysis in a realistic hydrodynamic framework. Please see also our explanation regarding the chosen complexity of our model above and L153. The zooplankton grazers are described to represent copepods. We will specify this in the revised version of the manuscript. Adding more taxonomic and trophic complexity such as described in Flynn et al. 2022, is an important angle for future research efforts. We will discuss these possible future directions in the revised version.

Fig.5 These 'future' plots carry even more caveats than does the control. All of these appears rather too much like a 'first try' rather than a comprehensive attempt to explore the dynamics.

Fig. 5 We would like to refer to our explanation regarding the chosen complexity of our model above and our description in the model biases. Since there are hardly any experimental, observational or modelling data available on the potential role of viruses under climate change, we believe our study adds valuable information on this topic.